# The determination of causality of drug induced liver injury in patients with COVID-19 clinical syndrome

Lina Mohammad Naseralallah[1,2], Bodoor Abdallah Aboujabal[1], Nejat Mohamed Geryo[3], Aisha Al Boinin[4], Fatima Al Hattab[4], Raza Akbar[3], Waseem Umer[3], Layla Abdul Jabbar[3], Mohammed I. Danjuma[3,4,5]*

1 Clinical Pharmacy Department, Hamad Medical Corporation, Doha, Qatar, 2 School of Pharmacy, College of Medical and Dental Sciences, University of Birmingham, Birmingham, United Kingdom, 3 Division of General Internal Medicine, Weill Cornell affiliated-Hamad General Hospital, Hamad Medical Corporation, Doha, Qatar, 4 College of Medicine, QU Health, Qatar University, Doha, Qatar, 5 Weill Cornell College of Medicine, New York, Doha, Qatar

* mdanjuma@hamad.qa

## Abstract

### Background

Drug induced liver injury (DILI) is a rising morbidity amongst patients with COVID-19 clinical syndrome. The updated RUCAM causality assessment scale is validated for use in the general population, but its utility for causality determination in cohorts of patients with COVID-19 and DILI remains uncertain.

### Methods

This retrospective study was comprised of COVID-19 patients presenting with suspected DILI to the emergency department of Weill Cornell medicine-affiliated Hamad General Hospital, Doha, Qatar. All cases that met the inclusion criteria were comparatively adjudicated by two independent rating pairs (2 clinical pharmacist and 2 physicians) utilizing the updated RUCAM scale to assess the likelihood of DILI.

### Results

A total of 72 patients (mean age 48.96 (SD ± 10.21) years) were examined for the determination of DILI causality. The majority had probability likelihood of "possible" or "probable" by the updated RUCAM scale. Azithromycin was the most commonly reported drug as a cause of DILI. The median R-ratio was 4.74 which correspond to a mixed liver injury phenotype. The overall Krippendorf's *kappa* was 0.52; with an intraclass correlation coefficient (ICC) of 0.79 (IQR 0.72–0.85). The proportion of exact pairwise agreement and disagreement between the rating pairs were 64.4%, *kappa* 0.269 (ICC 0.28 [0.18, 0.40]) and *kappa* 0.45 (ICC 0.43 [0.29–0.57]), respectively.

**Data Availability Statement:** All relevant data are within the paper.

**Funding:** The author(s) received no specific funding for this work.

**Competing interests:** The authors have declared that no competing interests exist.

## Conclusion

In a cohort of patients with COVID-19 clinical syndrome, we found the updated RUCAM scale to be useful in establishing "possible" or "probable" DILI likelihood as evident by the respective *kappa* values; this results if validated by larger sample sized studies will extend the clinical application of this universal tool for adjudication of DILI.

## Introduction

Drug-induced liver injury (DILI) is a, challenging and complex adverse event caused by exposure to certain medications [1, 2]. It is characterized by increase in serum alanine aminotransferase (ALT) and alkaline phosphatase (ALP) levels which are considered reliable markers of liver tissue damage. Patterns of change in these laboratory markers could be hepatocellular, cholestatic or mixed, depending on the type of liver injury [1]. The epidemiological burden of DILI is variable, with a recent retrospect report estimating it at about 1.91 events per million person-years with a corresponding mortality rate of 21% [3]. Establishing DILI-drug causality is often fraught with lots of diagnostic difficulties. The harmonization of DILI causality tools by the introduction of the Roussel Uclaf Causality Assessment Method (RUCAM) and its subsequent update have resolved some of the subsisting uncertainties [4]. This scale is a scoring algorithm that allows clinicians to assign points based on presence or absence of clinical, biochemical, serologic, and radiological features of liver injury [5–7]. It is an objective tool that has been able to establish causality between DILI-drug pairs [8]. Given the robustness and reliability of the updated RUCAM scale, several studies have utilized it, either prospectively or retrospectively, to identify DILI-drug pair(s) in different cohorts of patients [6, 9–12].

The novel coronavirus disease-2019 (COVID-19) clinical syndrome is a highly transmittable and pathogenic viral infection caused by the severe acute respiratory syndrome coronavirus 2 (SARS-CoV-2) [13]. A recent review showed that 46% of COVID-19 patients had elevated plasma aspartate aminotransferase (AST), 35% had elevated ALT with 2–5% of patients having elevated ALP [14]. Consequent upon this, recent reports are suggesting that abnormal LFTs are a common feature of COVID-19 clinical syndrome [13, 14]. Very often uncertainty regarding the exact therapeutic approach to addressing COVID-19 clinical infection meant that multiple cocktails of medications in varying combinations and permutations were used as suggested by hurriedly constituted national and international clinical guidelines. These includes suppressive antivirals, antibiotics, antiprotozoal, and immunosuppressants agents; all of which has the potential to cause various of phenotypes of liver biochemistry abnormalities [15]. When these patients experience abnormalities in their LFTs during the course of therapy, it then becomes difficult to establish causality of a probable DILI-drug pair with reasonable degree of certainty. To date, no published studies have utilized the updated RUCAM scale *ab initio* to investigate its performance in the determination of DILI in this challenging cohort of patients with vastly increased multiplicative risks of DILI. In this study, we have attempted a comparative causality determination of probable DILI in patients with the COVID-19 clinical syndrome with the view of determining the performance of the updated RUCAM scale in this cohort of patients.

## Materials and methods

This phase IV retrospective study that recruited all consecutive patients presenting to the Emergency Department of Weill Cornell Medicine-affiliated Hamad Medical Corporation,

Doha, Qatar with confirmed COVID-19 clinical syndrome and were suspected of having DILI during their treatment. Patients were eligible for enrollment in this study if they were 18 years of age or older, had confirmed diagnosis of COVID-19 clinical syndrome (through positive nasopharyngeal PCR swab), and were suspected of having DILI with complete biochemical and socio-demographic records available for analysis. Additionally, these patients underwent a thorough investigation to rule out alternative diagnoses (negative hepatitis serology [A, B, C, EBV, CMV], autoimmune screens, normal liver and biliary ultrasound, and no known exposure to hepatotoxic substances such as acute alcohol ingestion). We did not exclude any acetaminophen-induced DILI, however cases with acetaminophen overdose were excluded. Patients who failed to satisfy any of the inclusion criteria were excluded. Relevant socio-demographic and laboratory parameters were abstracted from an online patient information management system (Cerner) into a Microsoft Excel data collection spreadsheet. Data extracted includes age, gender, date of COVID-19 diagnosis, COVID-19 related medications, other medications, date of commencement and cessation of medications, results of investigations for deranged LFTs including hepatitis serologies (Hepatitis A, B, C) and liver ultrasound, and any re-challenge (where appropriate) and its results.

The updated RUCAM scale, a validated structured causality assessment tool was used for the causality adjudication process to determine the likelihood of DILI [4]. As there is no validated published manual for raters training before using the tool, a random sample of 5 DILI-drug pairs were selected for pilot testing and training by all adjudicators. Subsequently, two independent rating pairs (2 clinical pharmacists and 2 general physicians) determined the likelihood of DILI using the scale. Each independent rater initially estimated the R-value (ALT/ upper limit of normal (ULN) divided by ALP/ ULN). Cases with R-ratio >5 was established as being DILI of hepatocellular type, whereas those with R-score between 2–5 and <2 were classified as mixed and cholestatic, respectively. Subsequently, suspected DILI-drug pairs were scored with the updated RUCAM scale based on the affirmation of the classification. Final scores were interpreted as follows: ≤0 indicate that the drug is "excluded" as a cause of DILI; 1 to 2 indicates that DILI is "unlikely"; 3 to 5 "possible"; 6 to 8 "probable"; and >8 "highly probable" [7]. An ethical approval was obtained from the independent review board (IRP) of the Medical Research Centre (HMC).

## Statistical analysis

Continuous variables were presented as means (± standard deviation (SD), or median (interquartile range (IQR)) depending on distribution. DILI causality gradings were expressed as categorical variables, with their pairwise interrater agreement proportions, Krippendorf's *kappa* statistics with 95% confidence intervals (CI), and intraclass correlation coefficients (ICC). To determine agreement proportions across multiple assessors, we calculated and compared the exact pairwise scores with a global *kappa* score. Analyses were conducted using Real Statistics Resource Pack software (Release 7.6). Copyright (2013–2021) Charles Zaiontz. www. real-statistics.com

## Case definitions

- **DILI**: ALT or AST levels greater than 5 × the ULN and/or ALP level greater than 2 × the ULN on two consecutive occasions (at least two weeks apart) [7].

- **COVID-19 positive**: A positive result from a real-time reverse-transcription polymerase chain reaction (RT-PCR) from a nasopharyngeal swab [16].

- **Exact Agreement (EA)**: A situation where 2 raters scored the same DILI-drug pair to the same outcome (e.g. probable-probable).

- **Exact disagreement (ED)**: A situation where 2 raters scored the same DILI-drug pair to discordant outcomes (excluded-highly probable, excluded-probable, unlikely-highly probable, unlikely-probable).

- *Kappa* **values** of ≤ 0.20, 0.21–0.40, 0.41–0.60, 0.61–0.80, and 0.81–1 correspond to slight, fair, moderate, substantial, and almost perfect agreement, respectively [17].

- **Intraclass correlation coefficient (ICC) values** of < 0.5, 0.5–0.75, 0.75–0.9, >0.90 are indicative of poor reliability, moderate reliability, good reliability, and excellent reliability [18].

## Results

A total of 225 patients were screened for eligibility, of which 72 met the inclusion criteria and were enrolled in the study (Fig 1). Socio-demographic and baseline characteristics of study participants are shown in Table 1. The mean age of patient population was 46.6 (±SD 17.4) years, 36 (46.1%) of which were females. The macrolide antibiotic azithromycin accounted for a plurality of the DILI-drug pairs (33.3%), followed by 15.3% for both hydroxychloroquine (HCQ) and lopinavir (LPV) (Table 2). The median R-score for the study cohort was 4.74 (IQR 3.63, 6.86), suggestive of a mixed liver injury phenotype. The median RUCAM scale causality assessment value was 4 (IQR 5, 6). Table 3 illustrates the detailed likelihood outcomes based on the updated RUCAM scale final scores. We were not able to establish any significant correlation between age and the type of hepatic injury as the point estimate of age vis-à-vis liver injury phenotype appears uncertain ($P = 0.86$). However, we noted a significantly higher levels of ALT ($P = 0.01$) in patients with hepatocellular compared to the other biochemical phenotypes of DILI (Figs 2 and 3).

### Inter-rater agreement and reliability

Utilizing the updated RUCAM scale by the rating pairs resulted in a total of 288 decisions. The overall Krippendorf's *kappa* was 0.52, with an intraclass correlation coefficient (ICC) of 0.79 (IQR 0.72–0.85). This represents "excellent reliability" for utilizing the updated RUCAM scale. The average percentage pairwise agreement between the four rating pairs was 59.7% (Table 4 and Fig 4).

### Proportion of exact agreements and disagreements

The proportion of average exact pairwise agreement between the raters was 64.4%, *kappa* 0.269 (ICC 0.28 [0.18, 0.40]). The average pairwise Cohen's *kappa* was 0.45 (ICC 0.43 [0.29–0.57]). Table 4 gives the result of EA and ED amongst rating pairs.

## Discussion

To our knowledge, this is the first attempt at utilizing the updated RUCAM scale to ascertain causality of DILI-drug pairs in a population of patients with COVID-19 clinical syndrome. We found that most cases were rated with a "possible" and "probable" level of causality on the updated RUCAM scale (45.83% and 34.72%, respectively). This is supported by excellent inter-rater reliability (IRR) of 0.52. This is crucial as the use of multiple potentially hepatotoxic medications during COVID-19 treatment makes this population more prone to developing DILI and more challenging to adjudicate with certainty [15]. Indeed, a recent systematic review of

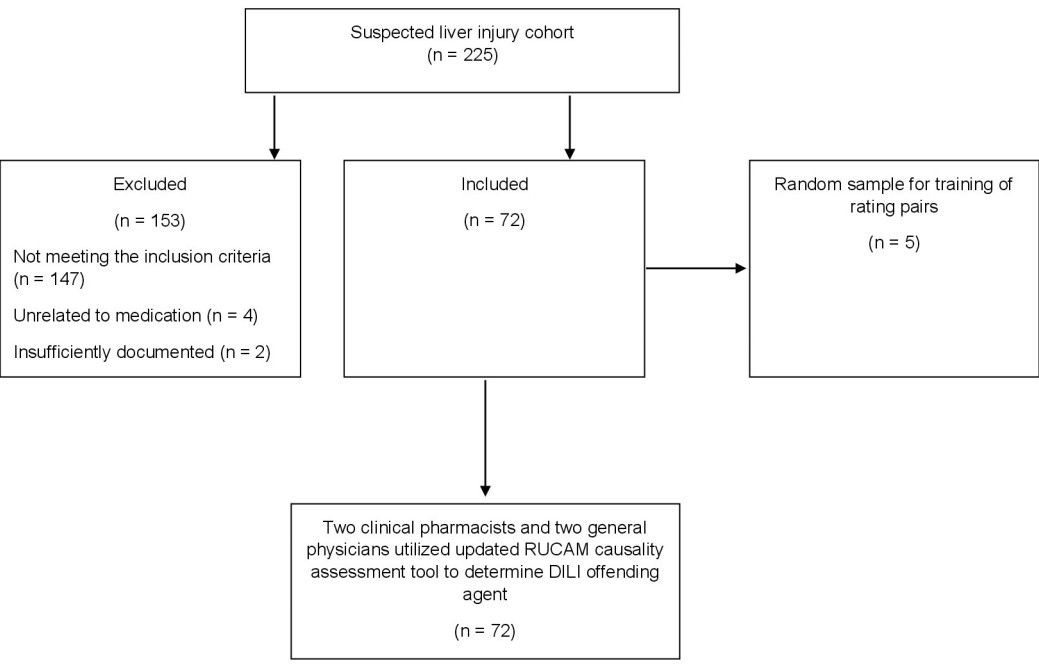

**Fig 1. Flow chart of participants recruitment.**

DILI in COVID-19 patients by Sodeifian et al. [19] reported medications (DILI) as a significant contributor of liver injury in these cohorts of patients, in addition to possibly the virus itself. This high proportion of agreement shows that utilization of RUCAM scale in COVID-19 cohort is robust and reliable, which aligns with the results reported by the original developers of the scale, and previous reports investigating its performance in different patient cohorts (including high risk population such as elderly) [6, 7, 9, 10]. Therefore, implementing the RUCAM scale in COVID-19 patients provide an objective and uniform approach for determining the likelihood of drug involvement, which is extremely important as early identification and discontinuation of the potential offending agent is the most critical component of the management process [20].

In almost all cases (91.66%), the culprit drug was an antimicrobial agent, which is expected given the natural history of the clinical syndrome (COVID-19) and the guideline-suggested

**Table 1. Demographic characteristics of the study population (n = 72).**

| Variable | N = 72 | Reference range |
|---|---|---|
| Age (years) Mean ± SD | 46.6 ± 17.4 | - |
| Gender (female) N (%) | 36 (46.1) | - |
| ALP (U/L) Median (IQR) | 98 (78–122) | 35–104 |
| ALT (U/L) Median (IQR) | 124 (90–195.8) | 7–56 |
| AST (U/L) Median (IQR) | 166 (134–225) | 5–40 |
| PT (sec) Median (IQR) | 11.05 (10.3–11.7) | 9.4–12.5 |
| DDM (mg/L FEU) Median (IQR) | 0.58 (0.285–1.23) | 0.00–0.49 |
| Ferritin (μg/mL) Median (IQR) | 1238 (279.75–2596.75) | 30–490 |

SD: Standard deviation; IQR: Interquartile range; ALP: Alkaline phosphatase; ALT: Alanine aminotransferase; AST: Aspartate aminotransferase; PT: Prothrombin time; DDM: D-Dimer

**Table 2. The Distribution of DILI implicated medications.**

| Medication | N (%) |
|---|---|
| Azithromycin | 24 (33.33) |
| HCQ | 11 (15.28) |
| LPV | 11 (15.28) |
| Ceftriaxone | 10 (13.89) |
| Paracetamol | 4 (5.56) |
| Amoxicillin-clavulanate | 3 (4.17) |
| Antifungal agent | 3 (4.17) |
| Cefuroxime | 2 (2.77) |
| Favipiravir | 2 (2.77) |
| Labetalol | 1 (1.39) |
| Statin | 1 (1.39) |

HCQ: Hydroxychloroquine; LPV: Lopinavir

treatment pathways. Additionally, our findings were consistent with those reported by Andrade eta al and Danjuma et al who reported from the General and elderly populations respectively [9, 10]. These reports add to the rising concerns around antimicrobial-induced hepatotoxicity [21]. It also speaks to the need for more post-marketing surveillance studies to assess the long-term outcomes and provide recommendations for the management of drug induced hepatotoxicity.

Our study design emphasis on incorporating clinical pharmacists and General physicians as the rating pairs for the determination of likelihood of DILI was deliberate; they represent the two broad groups of "shop floor" professionals that are most likely to encounter and be expected to provide clinical and therapeutic leadership in the event of apparent occurrence of a DILI-drug pair; additionally we wanted to test if an advanced knowledge of clinical pharmacology accounts for any significant variability on the ability to objectively use the scale. Our results demonstrated a nonsignificant difference between the two rating specialties (general physicians and clinical pharmacists) in the proportion of EA and ED. Physicians and pharmacists have shown comparable results in utilizing other diagnostic and causality determination tools, including the updated RUCAM scale amongst others [10, 22, 23]. This suggests that these tools are inherently objective, and do not require extensive knowledge of clinical pharmacology for their clinical application.

The main strength of our study is its novelty in been the first published attempt at exploring the utility of updated RUCAM scale in assessing the causality of DILI in patients with COVID-19 clinical syndrome. This will allow clinicians to make holistic therapeutic decisions as it will facilitate the prompt identification and cessation of suspected medications. Moreover, we

**Table 3. Final updated RUCAM causality assessment scale.**

| Likelihood | Frequency (N) | % |
|---|---|---|
| Excluded | 3 | 4.17 |
| Unlikely | 9 | 12.5 |
| Possible | 33 | 45.83 |
| Probable | 25 | 34.72 |
| Highly probable | 2 | 2.78 |

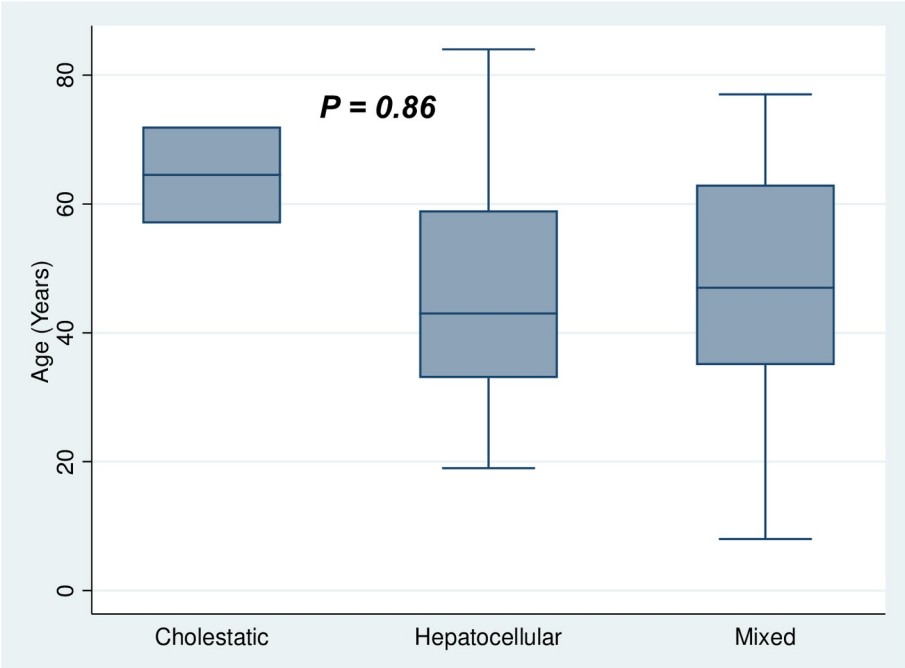

**Fig 2. The relationship between age and the different biochemical phenotypes of DILI.**

reported on the reassuring comparison between two rating specialties (general physicians and clinical pharmacologists) that are most likely to encounter DILI in their regular practice.

The interpretation of the findings of our study should be viewed in the context of the following limitations. By its retrospective design, the study was constrained by the usual issues

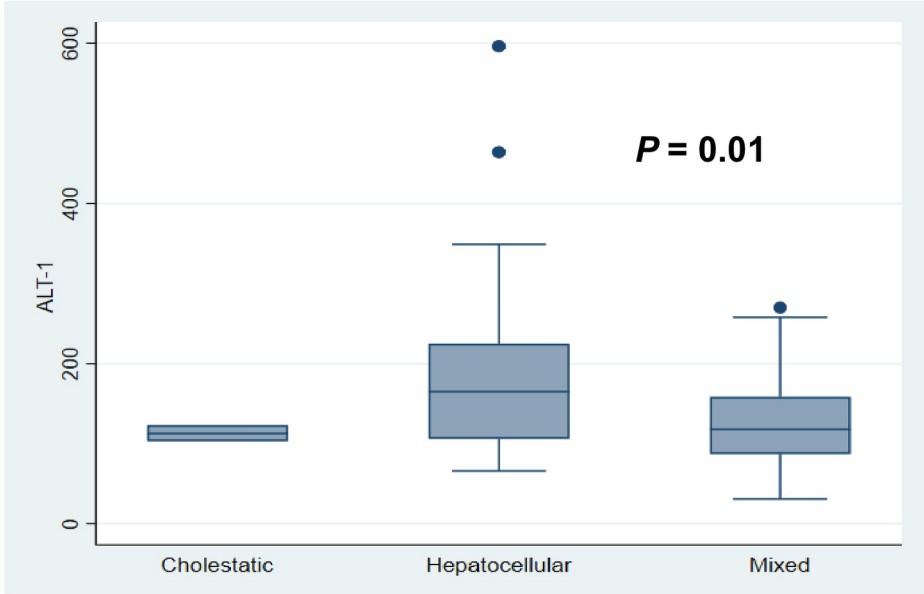

**Fig 3. Serum alanine aminotransferase (ALT) and the different biochemical phenotypes of DILI.**

**Table 4. Proportion of Cohen's *kappa* pairwise agreement/disagreement between rating pairs utilizing the updated RUCAM scale.**

| Rating pair | N | Pairwise EA | Pairwise ED |
|---|---|---|---|
| Raters 1&2 | 72 | 0.618 | 0.482 |
| Raters 1&3 | 72 | 0.535 | 0.397 |
| Raters 1&4 | 72 | 0.142 | 0.398 |
| Raters 2&3 | 72 | 0.644 | 0.636 |
| Raters 2&4 | 72 | 0.298 | 0.555 |
| Raters 3&4 | 72 | 0.1 | 0.684 |
| Total | **288** | **0.269** | **0.525** |

EA: Exact agreement; ED: Exact disagreement

associated with these data schemes. These includes missing data, absence of synchronized timing of LFT determination amongst others. As recently recommended by XX, a prospective study design is more likely to establish reliable causality of DILI-drug pairs with limited to negligible risks of significant confounding or impact on the constraints highlighted above. For our index study the emergency setting imposed by the COVID-19 pandemic makes a robust planning for prospective study design rather difficult. Additionally, there is no standardized published scheme for raters training before the use of the updated RUCAM scale. To overcome this, all raters were required to test run with 5 randomly selected DILI-drug pairs.

## Conclusion

In a cohort of patients with COVID-19 clinical syndrome, we found the updated RUCAM scale to be useful in establishing "possible" or "probable" DILI likelihood as evident by the respective *kappa* values; this results if validated by larger sample sized prospective cohorts will extend the clinical application of this universal tool for adjudication of DILI.

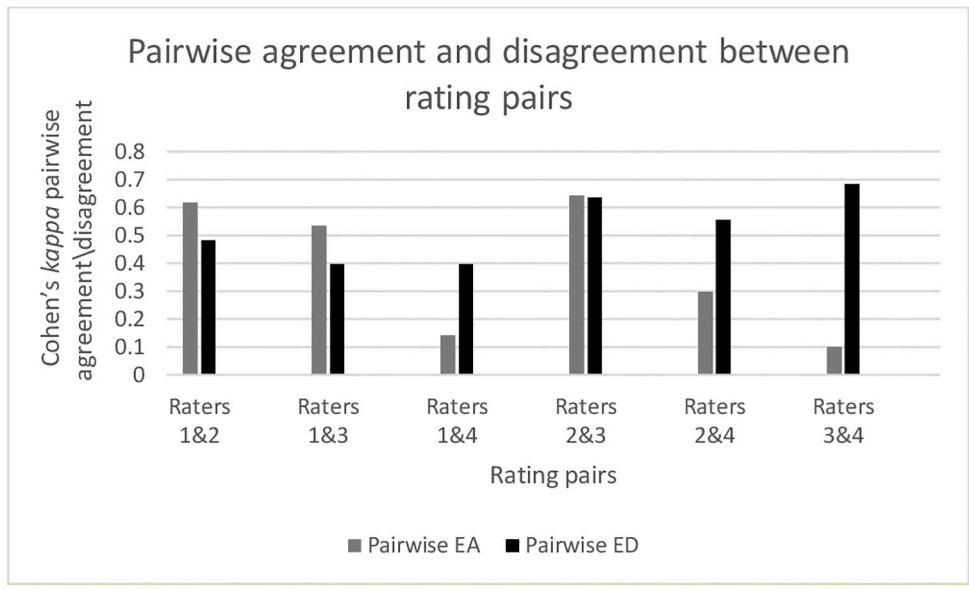

**Fig 4. Pairwise agreement and disagreement between rating pairs.**

## Author Contributions

**Conceptualization:** Lina Mohammad Naseralallah, Mohammed I. Danjuma.

**Data curation:** Lina Mohammad Naseralallah, Bodoor Abdallah Aboujabal, Nejat Mohamed Geryo, Aisha Al Boinin, Fatima Al Hattab, Raza Akbar, Waseem Umer, Mohammed I. Danjuma.

**Formal analysis:** Lina Mohammad Naseralallah, Bodoor Abdallah Aboujabal, Mohammed I. Danjuma.

**Funding acquisition:** Lina Mohammad Naseralallah, Mohammed I. Danjuma.

**Investigation:** Lina Mohammad Naseralallah, Bodoor Abdallah Aboujabal, Nejat Mohamed Geryo, Raza Akbar, Mohammed I. Danjuma.

**Methodology:** Lina Mohammad Naseralallah, Bodoor Abdallah Aboujabal, Mohammed I. Danjuma.

**Project administration:** Lina Mohammad Naseralallah, Mohammed I. Danjuma.

**Supervision:** Mohammed I. Danjuma.

**Validation:** Lina Mohammad Naseralallah, Bodoor Abdallah Aboujabal, Nejat Mohamed Geryo, Layla Abdul Jabbar, Mohammed I. Danjuma.

**Visualization:** Bodoor Abdallah Aboujabal, Mohammed I. Danjuma.

**Writing – original draft:** Lina Mohammad Naseralallah, Mohammed I. Danjuma.

**Writing – review & editing:** Lina Mohammad Naseralallah, Bodoor Abdallah Aboujabal, Nejat Mohamed Geryo, Aisha Al Boinin, Fatima Al Hattab, Raza Akbar, Waseem Umer, Layla Abdul Jabbar, Mohammed I. Danjuma.

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
