## [Decision Letter · Decision Letter 0]

10 Feb 2022

PONE-D-21-31507The determination of causality of drug induced liver injury in patients with COVID-19 clinical syndromePLOS ONE

Dear Dr. danjuma,

Thank you for submitting your manuscript to PLOS ONE. After careful consideration, we feel that it has merit but does not fully meet PLOS ONE’s publication criteria as it currently stands. Therefore, we invite you to submit a revised version of the manuscript that addresses the points raised during the review process.

We look forward to receiving your revised manuscript.

Kind regards,

Evy Yunihastuti, MD

Academic Editor

PLOS ONE

Journal Requirements:

Reviewers' comments:

Reviewer's Responses to Questions

**Comments to the Author**

1. Is the manuscript technically sound, and do the data support the conclusions?

Reviewer #1: Partly

Reviewer #2: Yes

2. Has the statistical analysis been performed appropriately and rigorously? 

Reviewer #1: No

Reviewer #2: Yes

3. Have the authors made all data underlying the findings in their manuscript fully available?

Reviewer #1: No

Reviewer #2: Yes

4. Is the manuscript presented in an intelligible fashion and written in standard English?

Reviewer #1: Yes

Reviewer #2: Yes

5. Review Comments to the Author

Reviewer #1: At first the mission of this study was very good and interesting because until now the diagnosis of DILI has always been a problem and is often placed at the end after other possible diagnoses have been ruled out. But upon further reading of this study, it was frankly very confusing. There were inconsistencies in the method because it was first said to be a retrospective cohort but later said to be cross sectional. Moreover, if this was done retrospectively, it would be very difficult to screen out DILI cases based on the RUCAM scale, which has many question components.Of course, there will be a lot of recall bias if the incident has passed. The cross sectional method is not suitable for determining DILI cases. Moreover, when the initial case of COVID occurred, by looking at the increase in the transaminases enzyme, it was very difficult to determine from the start whether the case was DILI or not.

Reviewer #2: The suggestions that we can give to the manuscript with the title "The determination of causality of drug induced liver injury in patients with COVID-19 clinical syndrome”, are :

1. The grammatical errors in the text were quite at large in number.

Authors need to send the manuscript for proofreading and correction.

2. In my opinion, some of the information provided in the Introduction section are more suitable to be included in the Discussion instead.

A lengthy introduction compared to the Discussion might give an impression that available studies in the literature on this matter is more than adequate in contributing the information without the need of the current study.

3. It is better if the authors can provide more updates or additional facts that can be obtained from this study compared to what already been published in the literature, and also presents data from others Asian countries, especially Southeast Asia, as comparison data.

4. As highlighted by the authors, study sample was very small, many patients were excluded.

5. I suggest that the authors display figures in a format that is easier to understand, especially table 4.

6. Reference : correct or cite in full reference numbers 7, 18, 19.

6. PLOS authors have the option to publish the peer review history of their article (what does this mean?). If published, this will include your full peer review and any attached files.

Reviewer #1: No

Reviewer #2: No

---

## [Author Response · Author response to Decision Letter 0]

17 Feb 2022

The editor-in-Chief 14/02/2022

Plos One

Dear sir,

The determination of causality of drug induced liver injury in patients with COVID-19 clinical syndrome 

Please find enclosed our response to reviewers’ comments. We believe that our response and changes made to the manuscript did enhance its scientific message and will be immensely useful to the readers of this journal

Kind regards,

Dr Mohammed I Danjuma

On behalf of other co-authors

Response to reviewer’s comment

Comment from the Editor 

“Please ensure that your manuscript meets PLOS ONE's style requirements, including those for file naming. The PLOS ONE style templates can be found at In general, the response is reasonable. However, several points require further clarification:”

Authors response

Editors comment noted with thanks. We have carried out an exhaustive review of the manuscript styling consistent with PLOS ONE editorial policy as advised.

Reviewers comment

“At first the mission of this study was very good and interesting because until now the diagnosis of DILI has always been a problem and is often placed at the end after other possible diagnoses have been ruled out. But upon further reading of this study, it was frankly very confusing. There were inconsistencies in the method because it was first said to be a retrospective cohort but later said to be cross sectional. Moreover, if this was done retrospectively, it would be very difficult to screen out DILI cases based on the RUCAM scale, which has many question components. Of course, there will be a lot of recall bias if the incident has passed. The cross-sectional method is not suitable for determining DILI cases. Moreover, when the initial case of COVID occurred, by looking at the increase in the transaminases enzyme, it was very difficult to determine from the start whether the case was DILI or not”

Authors response 

We have noted reviewers’ comment with thanks. We regret the oversight if there was an interchangeable mention of study design between “cross sectional” and “retrospective” in different sections of the manuscript. Our study design was retrospective and despite its obvious limitations (as ably highlighted by the reviewer), the changing pattern and clinical phenotypes of DILI across different clinical environment and patient settings, meant that a significant proportion of both the foundational and subsequent published work on it (with RUCAM as a causality determination tool) has been through a mixture of both prospective and retrospective study designs (1,2, 3, 4). We do recognize prospective design is more robust in DILI determination and less prone to confounding especially recall bias. That notwithstanding, peculiar circumstances especially in the setting of COVID-19 pandemic meant that the priority in the initial cycle of the pandemic for example was largely driven by the need to develop new therapeutics and prevention strategies with little or no focus on strategies to examine the effects of these therapeutics. Most of the data that recent studies on DILI are examining were the retrospectively collected repository data that accrued during the pandemic; and have since provided an excellent repository platform to both detect emerging “drug signals” but to also investigate various well validated pathways and algorithms which have been in clinical use (such as our study). But we do recognize the legion of retrospective study design limitations, and have acknowledged as much in the main manuscript (highlighted in yellow)

Reviewer’s comment

“The grammatical errors in the text were quite at large in number”

Authors response 

Reviewer’s comment noted with thanks. We regret the oversight. We have exhaustively reviewed the manuscript and have made relevant corrections to spelling mistakes and grammatical errors as advised

Reviewer’s comment

“In my opinion, some of the information provided in the Introduction section are more suitable to be included in the Discussion instead.

A lengthy introduction compared to the Discussion might give an impression that available studies in the literature on this matter is more than adequate in contributing the information without the need of the current study”.

Authors response 

Reviewer’s comment noted with thanks. We regret the oversight. We have re-written the introduction section to reflect the current understanding of uncertainties that still exists in the adjudication of DILI in patients with COVID-19 clinical syndrome.

Reviewer’s comment

“it is better if the authors can provide more updates or additional facts that can be obtained from this study compared to what already been published in the literature, and also presents data from others Asian countries, especially Southeast Asia, as comparison data” 

Authors response 

We have noted reviewers’ comment. We have already cited several studies and authorities from Asia and different parts of the world to provide a comparative context to this current work (reference 6 and 9-12). Additionally, DILI adjudication with RUCAM scale in patients with COVI-19 is an emerging challenge and will explain the relatively few studies there are for a robust comparison with ours. 

Reviewer’s comment

As highlighted by the authors, study sample was very small, many patients were excluded.

Authors response 

We note reviewers’ comments with thanks. We did acknowledge that one of the limitations of our work is its relatively small sample size for which we have already recommended validation by a larger sample sized prospective patient cohort (highlighted in yellow).

Reviewer’s comment

I suggest that the authors display figures in a format that is easier to understand, especially table 4.

Authors response 

Many thanks for the suggestion. We have additionally carried a pictorial representation of the data in table 4 in form of Bar chart

Reviewer’s comment

“Reference: correct or cite in full reference numbers 7, 18, 19.”

Authors response

We noted reviewers’ comment with thanks and regret the oversight regarding the highlighted references. We have amended those references as appropriate.

References 

1. Abid A, Subhani F, Kayani F, Awan S, Abid S. Drug induced liver injury is associated with high mortality-A study from a tertiary care hospital in Pakistan. PLoS One. 2020 Apr 10;15(4)

2. Chen F, Chen W, Chen J, et al. Clinical features and risk factors of COVID-19-associated liver injury and function: A retrospective analysis of 830 cases. Ann Hepatol. 2021; 21:100267

3. Danan G, Teschke R. Roussel Uclaf Causality Assessment Method for Drug-Induced Liver Injury: Present and Future. Front Pharmacol. 2019; 10:853. Published 2019 Jul 29. doi:10.3389/fphar.2019.00853

4. Cunningham M, Iafolla M, Kanjanapan Y, Cerocchi O, Butler M, Siu LL, et al. (2021) Evaluation of liver enzyme elevations and hepatotoxicity in patients treated with checkpoint inhibitor immunotherapy. PLoS ONE 16(6): e0253070

---

## [Decision Letter · Decision Letter 1]

6 May 2022

The determination of causality of drug induced liver injury in patients with COVID-19 clinical syndrome

PONE-D-21-31507R1

Dear Dr. danjuma,

We’re pleased to inform you that your manuscript has been judged scientifically suitable for publication and will be formally accepted for publication once it meets all outstanding technical requirements.

Kind regards,

Evy Yunihastuti, MD

Academic Editor

PLOS ONE

Additional Editor Comments (optional):

Reviewers' comments:

Reviewer's Responses to Questions

**Comments to the Author**

1. If the authors have adequately addressed your comments raised in a previous round of review and you feel that this manuscript is now acceptable for publication, you may indicate that here to bypass the “Comments to the Author” section, enter your conflict of interest statement in the “Confidential to Editor” section, and submit your "Accept" recommendation.

Reviewer #1: All comments have been addressed

Reviewer #2: All comments have been addressed

2. Is the manuscript technically sound, and do the data support the conclusions?

Reviewer #1: Yes

Reviewer #2: Yes

3. Has the statistical analysis been performed appropriately and rigorously? 

Reviewer #1: Yes

Reviewer #2: Yes

4. Have the authors made all data underlying the findings in their manuscript fully available?

Reviewer #1: Yes

Reviewer #2: Yes

5. Is the manuscript presented in an intelligible fashion and written in standard English?

Reviewer #1: Yes

Reviewer #2: Yes

6. Review Comments to the Author

Reviewer #1: The authors have responded appropriately. This study has a valuable and novelty idea. Some issues has been clarified clearly.

Reviewer #2: Suggestions for displaying research data in an easy-to-understand display, as well as the need for further research in many centers with heterogeneous samples.

7. PLOS authors have the option to publish the peer review history of their article (what does this mean?). If published, this will include your full peer review and any attached files.

Reviewer #1: **Yes: **Chyntia Olivia Maurine Jasirwan, MD, PhD

Reviewer #2: No

---

## [Editor Report · Acceptance letter]

11 Aug 2022

PONE-D-21-31507R1 

The determination of causality of drug induced liver injury in patients with COVID-19 clinical syndrome 

Dear Dr. Danjuma:

I'm pleased to inform you that your manuscript has been deemed suitable for publication in PLOS ONE. Congratulations! Your manuscript is now with our production department. 

Kind regards, 

on behalf of

Dr. Evy Yunihastuti 

Academic Editor

PLOS ONE